# Platelets, Protean Cells with All-Around Functions and Multifaceted Pharmacological Applications

**DOI:** 10.3390/ijms24054565

**Published:** 2023-02-26

**Authors:** Chiara Puricelli, Elena Boggio, Casimiro Luca Gigliotti, Ian Stoppa, Salvatore Sutti, Mara Giordano, Umberto Dianzani, Roberta Rolla

**Affiliations:** 1Department of Health Sciences, Università del Piemonte Orientale, Via Solaroli 17, 28100 Novara, Italy; 2Maggiore della Carità University Hospital, Corso Mazzini 18, 28100 Novara, Italy; 3NOVAICOS s.r.l.s, Via Amico Canobio 4/6, 28100 Novara, Italy

**Keywords:** platelets, inflammation, platelet derivatives, extracellular vesicles

## Abstract

Platelets, traditionally known for their roles in hemostasis and coagulation, are the most prevalent blood component after erythrocytes (150,000–400,000 platelets/μL in healthy humans). However, only 10,000 platelets/μL are needed for vessel wall repair and wound healing. Increased knowledge of the platelet’s role in hemostasis has led to many advances in understanding that they are crucial mediators in many other physiological processes, such as innate and adaptive immunity. Due to their multiple functions, platelet dysfunction is involved not only in thrombosis, mediating myocardial infarction, stroke, and venous thromboembolism, but also in several other disorders, such as tumors, autoimmune diseases, and neurodegenerative diseases. On the other hand, thanks to their multiple functions, nowadays platelets are therapeutic targets in different pathologies, in addition to atherothrombotic diseases; they can be used as an innovative drug delivery system, and their derivatives, such as platelet lysates and platelet extracellular vesicles (pEVs), can be useful in regenerative medicine and many other fields. The protean role of platelets, from the name of Proteus, a Greek mythological divinity who could take on different shapes or aspects, is precisely the focus of this review.

## 1. Introduction

The assumption that platelets are involved only in primary hemostasis and regulation of blood flow was put into question a long time ago, and now there is evidence that these cell fragments have a distinctive role in the immune response [1]. The description of platelet-leukocyte micro-aggregates dates back to the 1990s [2], and all nine Toll-like receptors (TLRs) described in humans are on platelets [3,4], supporting the hypothesis of platelet involvement at least in innate immunity. Moreover, platelets display Fcγ immunoglobulin G receptors (FcγR) [5,6] and CD40 ligand (CD40L), mediating the interaction with CD40 on B cells, dendritic cells, and macrophages, and they are able to sample the surrounding environment and present foreign pathogens and molecules to T cells [4,7]. All these attributes imply a bridge between innate and adaptive immunity so that platelets are not only involved in first protection against foreign antigens but also take part in the entire immune response in a much more comprehensive pattern.

When induced by pathogen-associated molecular patterns (PAMPs) during infections or by cell-damage-associated molecular patterns (DAMPs), TLRs can initiate intracellular signaling leading to platelet activation, to promote not only primary hemostasis but also the immune response. For instance, lipopolysaccharide (LPS)-mediated induction of TLR4 can promote the formation of platelet-neutrophil aggregates where platelets play a role as inducers of neutrophil extracellular traps (NETs). NETs consist of a network of histones, chromatin, and degradation enzymes released by neutrophils during a unique type of cell death (NETosis), and their main goal is pathogen entrapment and elimination through oxidative and non-oxidative mechanisms, thus limiting their diffusion in the bloodstream [1,8]. The complex formed by platelets plus NETs thus serves as a scaffold to bind and eliminate pathogens on one side and amplify platelet activation on the other [1]. Furthermore, the phenomenon of autophagy has been detected also in activated platelets [9]. Several platelet surface molecules and receptors have been associated with their ability to interact with immune cells, including P-selectin recognizing P-selectin glycoprotein ligand-1 (PSGL-1) on lymphocytes, neutrophils, and monocytes [1,10].

Even in the absence of foreign pathogens, platelets can take part in sterile inflammatory reactions underlying several pathological processes, such as the multi-step evolution of an atherosclerotic plaque.

### 1.1. Platelet Granules and Receptors

One of the most interesting features of platelets is the wide number of biologically active molecules contained in their granules. Platelets contain two main types of secretory organelles, α granules, and dense bodies (δ granules), and most effector functions depend on their secretion. As a consequence of granule fusion with the platelet plasma membrane, several granule molecules may be expressed on the platelet surface or released as soluble molecules (e.g., coagulation factors, mitogenic factors, angiogenic mediators, and chemokines) acting locally at sites of vascular injury or even systemically [1].

#### 1.1.1. α-Granules

Proteomic studies indicate that α-granules release more than 300 soluble proteins acting in processes such as blood coagulation, inflammation, immunity, cell adhesion and growth, and possibly other less-known activities [11]. First of all, α-granules contain many mediators of blood coagulation such as fibrinogen, von Willebrand factor (VWF), and adhesive proteins that mediate platelet-platelet and platelet-endothelial interactions. VWF of α-granules constitutes 20% of the total VWF protein, mainly in the high-molecular-weight forms [12]. Moreover, α-granules also contain components of the VWF receptor complex (GPIbα-IX-V), the main receptor of fibrinogen (integrin αIIbβ3), and the collagen receptor (GPVI) [13,14]. These receptors are constitutively expressed in resting platelets both on the plasma membrane and in α-granules (containing two-thirds of the whole αIIbβ3 and one-third of GPVI) and are upregulated on the membrane of activated platelets upon α-granule secretion.

The key molecules in α-granules are the basis of several coagulation pathways, secreted by activated platelets and involved in secondary hemostasis: Factor V, endocytosed from the plasma and stored in α-granules as activated factor V (FVa) complexed to the carrier protein; Factor XI and XIII, synthesized in megakaryocytes and stored in α-granules; Factor II prothrombin; high molecular weight kininogens (HMWK), involved in the intrinsic clotting cascade; and plasminogen activator inhibitor-1 (PAI-1) and α2 antiplasmin, which are protease inhibitors limiting plasmin-mediated fibrinolysis [15].

α-Granules may also contribute to hemostatic balance by releasing several proteins limiting the progression of coagulation: antithrombin III, inhibiting both the intrinsic and extrinsic pathways, C1-inhibitor, degrading plasma kallikrein, factor XIa, and factor XIIa, and protein S and tissue factor pathway inhibitor (TFPI). Moreover, they store the fibrinolytic proteinase plasmin and its inactive precursor plasminogen [15].

Activated platelets externalize the anionic phospholipid phosphatidylserine (PS) required to support all coagulation reactions, and produce P-selectin and CD40L (see below) stimulating monocyte production of tissue factor (TF), mostly bound to microvesicles, which bind and fuse with platelets to initiate coagulation on the platelet surface [16]. Therefore, platelets can contribute to both anti- and pro-coagulant activities, but it is not known whether these opposite functions are ascribable to distinct platelet subsets.

Moreover, α-granules contain several chemotactic factors, including platelet factor-4 (PF4), β-thromboglobulin, epithelial neutrophil-activating peptide 78 (ENA-78), growth-related oncogene-α (GROα), regulated upon activation, normal T-cell expressed and secreted chemokines (RANTES), monocyte chemotactic protein 1 (MCP-1), macrophage inflammatory protein 1α (MIP-1α), and thymus- and activation-regulated chemokine (TARC), playing key roles in inflammation by their ability to recruit and activate several types of leukocytes. PF4 induces neutrophil activation and β2-integrin-mediated adhesion to endothelial cells [17,18]. ENA-78 is a potent chemokine for neutrophils, GROα and MCP-1 for monocytes, RANTES for eosinophils, monocytes, and T lymphocytes, and MIP-1α for monocytes, T and B lymphocytes, NK cells, basophils, and eosinophils. TARC is a selective chemoattractant for T cell subsets expressing a class of receptors binding TARC with high affinity and specificity [19,20,21,22]. MIP-1α and TARC may play a role in atherosclerotic plaque destabilization and atopic dermatitis, respectively.

Besides chemoattractants, α-granules also store other types of immunomodulatory molecules, such as CD40L, triggering receptor expressed on myeloid cells-1 (TREM-1) ligand, and transforming growth factor (TGF)-β1. CD40L (CD154) is known as a costimulatory receptor expressed on activated T cells, binding to CD40, and expressed on the surfaces of B cells, endothelial cells, and dendritic cells. The CD40L/CD40 interaction plays a bidirectional role in the costimulation of both T cells and their CD40-expressing partner and in inducing immunoglobulin (Ig) isotype switching in activated B cells [1]. However, CD40L can also be produced in a soluble form (sCD40L) due to metalloproteinase-dependent cleavage of the membrane form or alternative splicing of the CD40L RNA. Besides CD40, CD40L can also bind to the integrins GPIIb/IIIa, α5β1 (CD49e/CD29), and αMβ2 (CD11b/CD18, Mac-1). Platelets are the main source of circulating sCD40L, which is involved in the inflammatory and prothrombotic responses: plasma levels of sCD40L are routinely used as a systemic marker of platelet activation. In several cohorts, plasma sCD40L predicts clinical cardiovascular adverse events. sCD40L levels are increased in plasma from patients with sickle cell disease, likely reflecting platelet activation [23]. TREM-1, a member of the V-type immunoglobulin superfamily, is constitutively expressed by neutrophils and monocytes [24]. Engagement of TREM-1 by its ligand expressed by platelets stimulates an oxygen burst and IL-8 production in neutrophils [25]. TGF-β is stored in large amounts in platelet α granules, and platelet storage seems to be important for maintaining circulating levels of this potent immunoregulatory factor.

All of the several α-granule mediators described above influence inflammation by exerting proinflammatory and immunomodulatory activities, mainly by recruiting and activating several leukocyte types. Platelets also contribute to the inflammatory process by expressing receptors that facilitate the adhesion of platelets to other cells. Most α-granule membrane-bound proteins are also present on the plasma membrane of resting platelets, including integrins (e.g., αIIb, α6, β3), immunoglobulin family receptors such asglycoprotein VI (GPVI), Fc receptors, platelet endothelial cell adhesion molecules (PECAMs), leucine-rich repeat family receptors (e.g., GPIb-IX-V complex), tetraspanins (e.g., CD9), and other receptors (CD36, Glut-3) [26]. Platelet activation induces upregulation of these molecules on the membranes of activated platelets. Several other α-granule membrane-associated proteins are not expressed on the surfaces of resting platelets and are expressed only upon activation, including the integral membrane proteins fibrocystin L, CD109, and P-selectin, so they are markers of activated platelets. In particular, P-selectin mediates platelet interaction with endothelial cells, monocytes, neutrophils, and lymphocytes by binding to PSGL-1, and it promotes their recruitment to sites of inflammation. Then, the platelet α-granule proteins fibrinogen, fibronectin, vitronectin, and VWF contribute to stabilize platelet-endothelial adhesion by forming cross-bridges between GPIIb-IIIa and the endothelial αVβ3 integrin or ICAM-1 [12].

#### 1.1.2. Dense Granules

Dense granules are nearly 10-fold less abundant than α-granules in human platelets. They contain many small molecules and comparatively fewer proteins. Among the former, key roles are played by ADP and ATP (650 mM and 440 mM, respectively), uracil and guanine nucleotides, calcium and potassium, and bioactive amines such as serotonin and histamine [27]. Platelet dense granules contain high concentrations of polyphosphates, whose release activates factor FXII and, in turn, the kallikrein-kinin system, resulting in the generation of bradykinin, inducing increased vascular permeability and edema in vivo [28,29]. The dense granule membrane proteins include CD63 (granulophysin) and LAMP-2. Several platelet plasma membrane proteins have also been identified in dense granule membranes, including GPIb and αIIbβ3.

Although platelets are traditionally recognized for their central role in hemostasis, the presence of chemotactic factors, chemokines, adhesion, and costimulatory molecules in their granules and membranes indicates that they may play an immunomodulatory role in the immune response besides their capacity to trigger blood coagulation and inflammation [30].

Therefore, platelets perform a sentinel role at the sites of vascular injury, which may be crucial in regulating not only the inflammatory response but also the adaptive immune response, taking part in pathogen clearance and tissue repair [1].

## 2. Platelet Derivatives

Platelets themselves possess therapeutic properties that have been exploited in recent years, especially in the field of regenerative medicine, and that show promising potential for the future. Moreover, increased life expectancy has made the aging population grow in the last few decades, and chronic disorders to become more and more common, so the regenerative potential of platelets and platelet derivatives has started to be investigated more deeply.

Platelet granules are a powerful source of platelet growth factors (PGFs), ranging from more platelet-specific molecules to bioactive mediators shared by other cells involved in the regenerative cascade (endothelial cells, macrophages, neutrophils, and fibroblasts). Their action is mediated by the interaction with tyrosine kinase receptors and subsequent activation of multifunctional intracellular signaling cascades [31,32,33]. A list of the main growth factors released by platelets upon their activation is shown in Table 1. Briefly, their broad functions can be summarized in four categories: (1) mitogenesis and cell differentiation; (2) chemotaxis and migration, with important implications in angiogenesis and re-epithelialization; (3) regulation of the inflammatory response; and (4) extracellular matrix (ECM) formation and remodeling [34,35,36,37,38,39,40,41,42,43].

With these premises, it is not surprising that transfusion and regenerative medicine have come to an agreement characterized by the advent of hemocomponents “not for transfusion use”, such as platelet concentrates (PCs), exploiting the properties of what has been addressed as the “secretome” of platelets [36]. PCs have been explored in several clinical fields, including wound healing, treatment of osteoarticular lesions, dry-eye syndrome, corneal ulcers, and many others. PCs are autologous or allogeneic platelet derivatives with a platelet concentration higher than the blood baseline (1.50–3.50 × 10^11^/L) [34]. Data are conflicting regarding the PC’s optimal platelet concentration, and it must be argued that the platelet count is not necessarily an index of efficacy unless platelets are also of good quality. Moreover, excessive concentrations of platelets may even be detrimental since very high concentrations of some PGFs, such as TGF-β, may have antiproliferative effects or induce counterproductive proteolysis in the ECM [44,45]. Weibrich et al. suggested an optimal concentration of around 1 × 10^6^/uL [46], which is now the working definition of platelet-rich plasma (PRP) [39]. In general, a PC platelet count that is four to five times higher than the basal count in whole blood seems to be the best compromise [35]. Platelet concentration is also influenced by the size of donor platelets and the donor hematocrit, since approximately 20% of platelets remain adsorbed in the red blood cells (RBC) pellet during PRP preparation [39].

PCs have been classified depending on their leukocyte, platelet, and fibrin content, and properties such as gelification, processing techniques, and possible applications. The main terms used include pure platelet-rich plasma (P-PRP), leukocyte and platelet-rich plasma (L-PRP), pure platelet-rich fibrin (P-PRF) and leukocyte and platelet-rich fibrin (L-PRF), platelet gel (PG), platelet lysate (PL), and, when PCs are used in the form of collyrium, serum eye drops (E-S), and PRP eye drops (E-PRP) [34,35,37,40,47,48]. The main advantage derived from their use is that, instead of single recombinant growth factors, PCs offer a broad range of bioactive molecules often acting synergistically and improving the overall treatment efficacy [35,40].

The processing steps and the peculiar characteristics of each PC are presented in detail in Table 2, while Figure 1 shows a series of images exemplifying the technical preparation of P-PRP, L-PRP, and PRF underlying the differences between the three.

The choice between autologous or donor-derived PCs mainly depends on ethical and safety issues as well as patient-related conditions. Besides being better-accepted by patients, autologous PCs are devoid of the risk of infection related to contamination of blood products. On the other hand, autologous PCs suffer from large variability in the quality of platelets since patients frequently have underlying comorbidities compared with healthy donors whose blood components are also prepared through standardized procedures [37]. Moreover, autologous PC use might not be feasible if the patient is being treated with drugs affecting platelet function that cannot be suspended, such as aspirin or other anti-aggregating agents often used in older subjects with underlying cardiovascular disorders [63].

### 2.1. Examples of Medical Applications of PCs

Autologous PRP was first used in cardiac surgery by Ferrari et al. in 1987, but at that time only to avoid the use of homologous blood products during intraoperative blood salvage techniques [64]. Cardiac surgery was again a field of application in the early 1990s as a fibrin sealant [65,66]. Most studies have been conducted in the fields of maxillofacial surgery, dentistry [53,54], wound care [67,68], dermatology and esthetic medicine [69], orthopedics [70], ophthalmology [71], and neurology [72,73] (Figure 2).

Chronic wound healing was probably one of the first “atypical” applications of platelets as a treatment. While acute wounds restore the anatomical and functional integrity of the lesion in a relatively short time and well-organized process, chronic and complicated wounds deviate from the traditional healing cascade, resulting in chronic inflammation and incomplete or prolonged reconstitution of the lesion’s integrity due to disturbing local and/or systemic factors such as concomitant infection, diabetes, underlying malignancies, and chronic inflammatory disorders [74].

Platelets participate in wound healing from the very beginning of the four-step process (coagulation-inflammation-proliferation-remodeling [74]), i.e., from the formation of the platelet plug, representing the first step in the hemostatic cascade. However, their role is not limited to blood loss prevention. Instead, the release of platelet granule content represents a key step influencing whole-tissue remodeling. In the first few days of wound healing, platelets modulate inflammatory events at the wound site mainly by promoting leukocyte chemotaxis. Angiogenesis, fibroblast proliferation, and ECM deposition through collagen synthesis and regulation of collagenase production characterize the next steps, and it is especially in this scenario that PGFs play a major role. The first clinical application of PRP was in 1991 in the setting of chronic leg ulcers, where the new formation of vascularized connective tissue could be demonstrated [75]. Since then, several trials have been conducted on human subjects, and a 2011 meta-analysis fully investigating the results obtained in the previous ten years agreed on the key role of PRP in accelerating the healing process, in both difficult-to-treat wounds and chronic ulcers such as those affecting diabetic patients. Furthermore, it seems that PRP also bears antimicrobial activity, which certainly accelerates the healing process by resisting local infections [76].

Just as in wound healing, bone repair also implies a tripartite process consisting of inflammation, proliferation, and remodeling. Here, the synergistic effect of platelet-derived TGF-β, PDGF, EGF, and b-FGF creates the most adequate microenvironment to sustain bone healing. In particular, they are primarily involved in osteoinduction by promoting differentiation of osteogenic precursors and mitogenesis, and, when bone grafts are used, they favor the process of osteoconduction by promoting vascular and cellular migration along a proper scaffold [37]. Starting from one of the most common orthopedic complaints, especially of older patients, i.e., osteoarthritis, PRP can support a cost-effective, low-intensity but high-quality therapeutic program where PRP shows beneficial effects on joint physiology, including chondrocyte proliferation, ECM production, and suppressed catabolism as well as an anti-inflammatory effect [70]. In addition, the osteoinductive properties of PCs have extended their applications beyond the field of orthopedics [77], broadly covering also maxillofacial surgery and dentistry [53,54,78]. Indeed, the osteoinductive properties of platelet derivatives improve the integration of dental implants, providing better grafting results.

The versatility of PCs has allowed the introduction of collyrium preparations that have proved to be beneficial in the treatment of ocular surface diseases such as dry eye syndrome or corneal ulcers [71], and even macular holes [79]. E-PRP functions as a lubricant and was shown to decrease inflammation in patients affected by disorders of the tear film and to accelerate the healing of corneal ulcers, and it is likely a promising alternative to topical steroids, which can quickly relieve symptoms, but whose long-term usage may result in side effects such as an increased risk of infection, cataract formation, and increased intraocular pressure [71]. Furthermore, compared to pure autologous serum, whose usage dates back to the 1980s [80], E-PRP seems to provide a slightly higher benefit, probably due to its platelet content, which allows prolonged release of growth factors ensuring a longer-lasting effect [71].

In addition to their regenerative potential, platelet derivatives may have analgesic properties as well. Data on the topic are conflicting [81,82,83,84,85,86], but studies showing a benefit in pain control suggest that the underlying mechanisms might be either the release of dense granule-derived serotonin, a well-known anti-nociceptive mediator in pain pathways, or an indirect analgesic effect due to the enhancement of nerve repair and re-myelination, thus eliminating the source of neuropathic pain [37,84,85].

A discussion on all the other possible clinical applications of PCs is beyond the scope of this review. However, Appendix A shows a complete list of the interventional and observational studies conducted since 1999. These results have been obtained after a search on clinicaltrials.gov using the keywords “platelet gel”, “platelet lysate”, “platelet derivatives”, “platelet-rich plasma”, “platelet-rich fibrin”, “serum eyedrop”, and “platelet-rich plasma eyedrop”. A total of 920 interventional and 41 observational studies have been identified, mostly in the fields of orthopedics, dentistry, and maxillofacial surgery.

### 2.2. Limitations in the Use of PCs

The greatest hindrance to the use of PCs is the scarce harmonization among studies, already starting from the terminology used to address the different platelet derivatives and encompassing a lack of standardization in preparation protocols, which may deeply influence the product’s quality in terms of platelet count, growth factor type and concentration in the final product, and, eventually, therapeutic efficacy [47,48]. An important issue is the centrifugation protocol, since high centrifugation forces may prematurely activate platelets resulting in a low final content of PGFs. The centrifugation speeds mentioned in the literature range from 180 g to 3000 g for 10–15 min depending on the platelet derivative to be prepared [37,39], and nowadays most procedures are still homemade and based on a “trial and error” approach. Comparison of results is thus difficult due to a lack of reproducibility and quality controls [87], and many open questions still remain in this intriguing field, ones regarding not only technical aspects but also the effective benefits that platelet derivatives could provide to patients in terms of healing efficacy, pain management, and anti-inflammatory effects. Even more, autologous donations require close logistic planning, and not all patients may be good candidates for this kind of approach, such as the cases of heavy smokers, cancer patients, or subjects taking anti-platelet drugs, whose platelet function is too altered to be of therapeutic relevance [63]. However, platelet extracellular vesicles (pEVs) might represent a promising alternative to overcome many limitations of PCs (see later).

## 3. Platelet Role in Disease

### 3.1. Platelets and Atherosclerosis

It is currently well-established that atherosclerosis is an inflammatory disease whose development largely depends on endothelial dysfunction. Rather than an anatomical disruption of the vessel wall, the starting point of atherogenesis is the intrinsic alteration of endothelial cell function, which in turn may be caused by many risk factors, including shear stress-related injury.

The formation of the atherosclerotic plaque involves several steps, starting from endothelial damage, and subsequently encompassing leukocyte transmigration through the vessel wall and proinflammatory cytokine production. In this process, the innate immunity contributes through macrophages that engulf oxidized low-density lipoprotein (LDL) particles, thus accumulating foam-like lipid droplets in their cytoplasm, turning into foam cells, and triggering a sterile proinflammatory intracellular signaling cascade. The involvement of the adaptive cell-mediated immune response through the major histocompatibility complex (MHC)-II-mediated antigen presentation to T helper cells completes the process and ensures an ongoing amplification of inflammation [88,89].

In this complex scenario, the contribution of platelets in atherosclerosis has been known since Fitzgerald et al. first reported increased thromboxane A2 levels as an index of platelet activation in unstable coronary artery disease in 1986 [90]. Platelet activation is likely the result of multiple contributing factors that, however, can be summarized with, again, the concept of endothelial dysfunction. Indeed, dysfunctional endothelial cells lose their anti-thrombotic properties (nitric oxide release, prostacyclin synthesis, or ecto-ADPase CD39 expression) and acquire a pro-thrombotic phenotype characterized by increased expression of tissue factor (TF), adhesion molecules, chemokines, and proinflammatory cytokines that eventually promote platelet tethering to an intact but dysfunctional vessel wall. In this way, even in the absence of a true break that needs to be repaired, the hemostatic cascade is triggered, starting from primary hemostasis, where platelets play a pivotal role.

Among the molecules involved in the platelet contribution to the atherosclerotic process, P-selectin appears to be indispensable, since it not only mediates platelet adhesion to the endothelium but also induces platelet-leukocyte aggregates to interact with the endothelium. Leukocytes, especially monocytes, express PSGL-1, and P-selectin binding to this receptor activates nuclear factor-κB (NFκB) and induces the expression of monocyte chemotactic protein-1 (MCP-1) together with a wealth of proinflammatory cytokines, primarily tumor necrosis factor α (TNF-α). In turn, platelet activation also leads to their release of cytokines synthesized de novo, such as interleukin-1β (IL-1β), and pre-formed molecules stored in platelet granules, especially α-granules, such as CD40L, the chemokine RANTES, MIP-1α, and PF4, which is crucial in promoting the uptake and the esterification of oxidized LDL by macrophages and their transformation into foam cells. On the whole, platelets are really at the core of atherogenesis, and they may be seen as facilitators and leukocyte chaperones at the atherosclerotic site. In the already-formed atherosclerotic plaque, they tend to form an adherent monolayer, which might be the forerunner of a future thrombotic process. Moreover, even when they adhere sparsely to the dysfunctional endothelium in the initial stages of atherogenesis, they can still mediate the delivery of proinflammatory and chemotactic factors or facilitate leukocyte-endothelium interactions, thus initiating and/or amplifying the whole process [89].

Beyond these initial stages, platelets also contribute to the subsequent remodeling of the atherosclerotic microenvironment characterized by vascular smooth muscle cell migration and proliferation and arterial intimal hyperplasia. In fact, α-granules release PDGF with chemotactic and mitogenic properties. Therefore, platelets contribute to the instability of the plaque and are directly involved in the sites of rupture of the atherosclerotic plaque, where they can cause, in situ, occlusion of the vessel and/or formation of thrombi that can occlude distant vessels [89].

It is therefore not surprising that the efficacy of antiplatelet agents such as aspirin might be related to a wide effect, perhaps including inhibition of the proinflammatory activity of platelets. Indeed, the antiplatelet treatment may affect the platelet-mediated inflammatory cascade in addition to playing a role in preventing aggregation and subsequent thrombotic events at the site of unstable plaques [91].

Acetylsalicylic acid (ASA) inhibits the enzyme cyclooxygenase (COX), of which two isoforms exist: COX-1 and COX-2. COX-1 is expressed constitutively in all tissues, while COX-2 expression is induced in inflammatory states. ASA can inhibit COX-1 by acetylating a serine residue at position 529 (Ser 529), and its antiaggregant effect is due to the reduction of thromboxane A2, a potent vasoconstrictor and inducer of platelet aggregation, whose production is COX-1-dependent. In contrast, ASA is 170 times less potent against COX-2 (through acetylation of Ser516), which plays a major role in inflammation by mediating the conversion of arachidonic acid into prostaglandin H2. As a result, the low doses of ASA usually recommended for cardiovascular prevention achieve a sufficient antiplatelet effect but an inadequate anti-inflammatory effect. Indeed, ASA has been shown to have greatest anti-inflammatory effects at doses above 1.2 g [92].

Among the other antiaggregant agents, thienopyridines such as clopidogrel have been extensively studied for their role extending beyond the antagonism of platelet aggregation. Different anti-inflammatory mechanisms have been proposed for this class of drugs, including decreased expression of adhesion molecules, chemokines, and TF, and a reduction in leukocyte aggregates formation [93]. These drugs antagonize the ADP P2Y12 receptor by irreversibly binding to a cysteine residue. Since ADP plays an essential role in platelet activation, the main result is an anti-aggregating effect. However, by doing so, thienopyridines also indirectly reduce platelet-mediated inflammation and interaction with leukocytes, for instance by reducing the P-selectin expression [94]. In addition, unlike ASA, which does not have any effect on leukocyte-platelet aggregates [95], thienopyridines can attenuate leukocyte function directly since the P2Y12 receptor has been found also on leukocytes [96].

Always dealing with the remodeling of the vascular microenvironment, another scenario where platelets are pathologically involved is the local evolution of vascular stents used to treat acute ischemic events. Basically, there are two main types of stents: bare metal stents and drug-eluting stents, with the latter being superior due to their release of antiproliferative drugs such as sirolimus, paclitaxel, or rapamycin, which prevent smooth muscle cell hyperplasia and hence restenosis [97,98]. On the other hand, they may also seriously injure the endothelium, contributing to the vicious cycle of endothelial dysfunction-platelet aggregation that has already been described in the pathogenesis of atherosclerosis. The consequence is a high risk of thrombosis and delayed endothelialization of the stent surface [99] and it is thus not surprising that antiplatelet drugs belong in the current therapy of patients receiving stent implants despite carrying an increased bleeding risk [100]. In this sense, several attempts have been made to optimize the engineering of stents in order to solve the urgent problem of obtaining biomaterials with antithrombotic, antiproliferative, anti-inflammatory, and pro-endothelialization properties within the same device [101]. The detailed achievements are beyond the scope of this narrative review. However, just to cite a few examples from the biomedical engineering field, Han and colleagues have developed a new coating of cardiovascular stents where the traditionally used magnesium alloy is combined with plant-derived ferulic acid, a highly biocompatible natural compound with anti-inflammatory properties and able to inhibit platelet aggregation and smooth muscle cell hyperplasia and to promote endothelial cell proliferation [102]. Even more interestingly, Li et al. have proposed a stent coated with a new subtype of chondroitin sulfate that, in addition to retaining the already-known anti-aggregating properties, shows what the authors call a “spatiotemporal orderliness of function”, i.e., a process in which the biomaterials direct cell fates in time and space sequence by influencing the surrounding microenvironment and inducing phenotype changes not only in vascular wall cells (smooth muscle and endothelial cells) but also in inflammatory cells, such as M1 and M2 macrophages [101,103].

These approaches demonstrate that a mindful choice of the biomaterials used in intravascular devices is of paramount importance for the patient outcome, and it must consider several variables regarding the different cell types involved in the complex process of device tolerance, including platelets.

### 3.2. Platelets and Cancer

Platelets are also involved in cancer, by regulating several aspects of tumorigenesis and metastasis. The interaction between cancer cells and platelets is very complex and bidirectional, both in the blood and in the tumor microenvironment.

Platelets contain more than 300 bioactive molecules in their granules (e.g., chemokines, platelet-derived growth factors) and express numerous receptors on their surfaces (e.g., P-selectin, integrin αIIbβIII, P2Y12, protease-activated receptor-1 (PAR-1)) directly involved in inflammation, cancer progression, and metastasis.

First, platelets are involved in the outcome of cancer patients. Elevated platelet counts are significantly correlated with a worse progression and lower overall survival in many cancers (breast, colon, lung, kidney, and pancreatic cancers). One mechanism that could explain thrombocytosis in many tumors could be tumor-derived interleukin-6 (IL-6), which stimulates thrombopoietin (TPO) production in the liver, thereby promoting megakaryopoiesis and thrombocytosis [104]. Moreover, a high percentage of cancer patients suffer from vascular thromboembolism (pulmonary embolism and deep venous thrombosis). Armand Trousseau first described, writing in 1865, that cancers can induce venous thrombus formation. Thrombosis is one of the most common clinical manifestations in cancer patients and is associated with worse prognosis and survival: the principal cause of high thrombotic risk is platelet activation and aggregation through direct and indirect mechanisms induced by tumor cells [104]. An important mechanism of tumor-induced platelet aggregation is the secretion by cancer cells of thrombin, a serine protease that converts fibrinogen to fibrin, activates many coagulation factors (FV, FVIII, FXI, and FXIII), and activates platelets by PAR. TF, the main activator of the coagulation cascade, is also expressed by cancer cells. High serum levels of TF have been found in several types of cancer. Activated platelets also express TF on their membrane, contributing to thrombosis [105]. Another mechanism of platelet activation by cancer cells is mediated by ADAM9 (a disintegrin and metalloproteinase 9), which is found in several cancer types and has been correlated with tumor aggressiveness and poor prognosis. ADAM9 can bind to the platelet laminin receptor (α6β1), a key platelet receptor for laminins. In this way, it supports both the adhesion and the activation of platelets and enhances platelet activation and tumor cell extravasation [106].

Secondly, platelets are directly involved in the progression and evolution of the tumor, playing an important role in metastases, whose formation is strongly inhibited by the cytotoxic activity of natural killer (NK) cells against the circulating tumor cells. Tumor cells and platelets are able to produce microthrombi in the circulation by the binding of the platelet P-selectin to the ligands of tumor P-selectin and thanks to the interaction between platelet TLR4 and the released High Mobility Group Box 1 (HMGB1) protein from the tumor [106]. The interaction between cancer cells and platelets is also mediated by the binding of GPIb-IX-V and GPIIb-IIIa to tumor cell integrin αvβ3 and via P-selectin, which can bind to mucins on the tumor cell membranes. Moreover, platelet-derived TGF-β diminishes NK cell activity by downregulating the NK activatory receptor NKG2D and increases tumor cell survival by activating the TGF-β/Smad and NF-kB pathways [104]. In this way, the metastases present in the circulation are protected/shielded by the platelets in a platelet aggregate, evading the immune response by their natural killers [107].

The activation of platelets by cancer cells and the subsequent “release reaction” of platelet secretome has many other pro-cancerous effects that modulate the tumor microenvironment, stimulate tumor growth and help metastases. After platelet activation, the platelet secretome is sequestered into the local microenvironment, where it can promote and support tumor cell proliferation and tumor growth. Activated platelets also release microvesicles [108] that can further promote disease progression through multiple mechanisms.

For example, to disseminate, circulating cancer cells need to adhere to endothelial cells and must infiltrate through the vessel wall. ATP secreted from dense granules activates P2Y2 on endothelial cells, increases endothelial permeability and promotes the diapedesis of cancer cells. In this way, platelets are directly involved in increasing endothelial permeability, promoting tumor cell transendothelial migration. The significant crosstalk between tumor cells and the endothelium, mediated by platelets, may then improve tumor metastasis.

Angiogenesis is another essential process in tumor growth and metastasis and, indeed, angiogenesis-based targeted therapy is considered a cornerstone for cancer treatment. PDGF and VEGF, secreted from *α*-granules, promote, respectively, tumor growth and angiogenesis [104].

Moreover, the proinflammatory cytokines released by platelets are powerful recruiters and activators of leukocytes; IL-8 and chemokines secreted by platelets attract hematopoietic cells to the tumor site, stimulating tumor growth and angiogenesis [109].

In this way, platelets participate directly in several steps of cancer metastasis and affect disease burden and treatment efficacy in cancer patients (Figure 3).

The strong evidence of an association between platelets and cancer, described above, has led to the hypothesis that antiplatelet therapy could be used in antitumor therapeutic strategies. It has been observed that, in adenoma, platelets that are activated by COX-2-mediated signal transduction pathways increase the aggressive phenotype of cancer cells. Low-dose aspirin, by inhibiting COX-1, may exert antimetastatic effects, decreasing the cancer-mediated activation and aggregation of platelets [110].

In a pancreatic tumor mouse model, daily administration of the antiplatelet drug clopidogrel decreased tumor growth rate and the number of metastatic foci significantly [111]. Similarly, the administration of antiplatelet drugs that bind glycoprotein complex αIIbβIII reduced the proliferation of melanoma cells injected in rat subcutis [107].

Platelets also play a role in protecting cancer cells against chemotherapy-induced apoptosis and in maintaining the integrity of tumor vasculature. In colon and ovarian cancer cell lines, platelets increase the resistance to 5-fluorouracil and paclitaxel. Moreover, low blood platelet counts in mouse models of breast and lung cancer significantly increase sensitivity to paclitaxel. In cancer patients and murine tumor models, high platelet counts have been associated with a poor response to chemotherapy [104].

Because of these pro-tumor effects, anti-platelet drugs have been introduced into cancer treatment strategies. Many pieces of evidence indicate that aspirin is useful in cancer to reduce metastasis and mortality, especially in colorectal cancer [106]. In 2016, the US Preventive Services Task Force approved the prophylactic use of low-dose aspirin in colorectal cancer patients [112].

Several other antiplatelet drugs are in the preclinical stage or are being tested in clinical trials, such as GPVI and GPIba antagonists [113]. GPVI is a platelet receptor for collagen on the subendothelial matrix and can bind also to fibrinogen and fibrin. GPVI promotes metastasis in mice [114] and may be involved in “tumor cell-induced platelet aggregation”. GPVI antagonists have broad inhibitory effects on tumor–platelet aggregation. Xu and colleagues observed that anti-GPIba antibodies decrease thrombopoietin generation and inhibit tumor-induced thrombocytosis [115].

Other antiplatelet agents can have antitumor effects, but their use in cancer therapy can be limited by their interference with hemostasis, increasing the risks of bleeding and gastrointestinal toxicity. In the near future, however, it will be important to further characterize the mechanism of action of antiplatelet agents in tumors, because it would certainly improve the antitumor therapeutic strategies.

### 3.3. Platelets, the Brain, and Neurodegenerative Conditions

The brain has the ability to reorganize itself throughout life to adapt to environmental changes via the continuous generation of new functional neurons derived from neural precursor cells. This process occurs in specialized neurogenic niches, predominantly in the subgranular zone of the hippocampal dentate gyrus and in the subventricular zone of the lateral ventricles [116].

The brain is well vascularized by a dense network of fine microvasculature, and molecular exchanges between the blood and the nervous system are finely controlled and influence neurogenesis during life.

Platelets are a link between the blood and the brain and can promote and modulate neurogenesis by secreting bioactive molecules from granules. Platelets can also rapidly respond to environmental changes in the brain by modifying their proteome via translation of stable mRNAs [116]. It has been recently demonstrated that platelets can cross the inflamed brain microcapillaries [117] and exert local actions through their surface receptors and released factors. Platelets express several surface molecules, which allow them to directly interact with glial cells, endothelial cells, and neurons [118]. In particular, CD62P, ALCAM, Siglec-H, and Siglec-15 platelet receptors can bind sialylated gangliosides present in the lipid rafts of neuronal processes [119]. These bindings would promote the formation of new dendritic spines on neurons and neural precursor cells [120] (Figure 4).

Platelets carry several neurotransmitters that are essential for the communication between neurons. In particular, they can promote and modulate neurogenesis by influencing neural precursor cells, and they can have neuroprotective effects via α-granules bioactive molecules, such as VEGF, EGF, FGF-2, IGF-1, PF4, TGF-β, and stromal cell derived factor 1 (SDF-1), and via dense granule neurogenesis-promoting molecules, such as serotonin, histamine, epinephrine, and dopamine [121] (Figure 4). It must be emphasized that platelets and neural cells are comparable in their intracellular storage compartments: platelet dense granules and small dense-core synaptic vesicles of neurons store serotonin and adenosine triphosphate contents, whereas the large dense-core vesicles of neurons and platelet α-granules contain neuropeptides, neurohormones, and neurotransmitters.

PRP treatment enhances the recovery of peripheral nerves following injury, including cavernous nerve injuries [121] and damage to the facial and sciatic nerves [122]. Moreover, PRP injections into the injured spinal cord of rats have been shown to promote locomotor recovery, local angiogenesis, and neuronal regeneration [123]. Another study in mice suggested the therapeutic use of PRP in neuroinflammatory central nervous system diseases [124].

In rats, PL injection promotes proliferation, neurogenesis, and survival and reduces apoptosis in neural precursor cells of the subventricular zone of the lateral ventricles [116]. Moreover, platelet-derived serotonin increases the expression of genes involved in synaptic plasticity [120]. Platelet microparticles and platelet exosomes can promote neural precursor cell proliferation, survival, and differentiation in vitro [125].

Human platelet lysates have been investigated as a novel biotherapy for amyotrophic lateral sclerosis (ALS) and Parkinson’s disease (PD) patients. In a cell-based model of ALS, human platelet lysates confer a neuroprotective effect against apoptosis and oxidative stress, inhibiting neuronal loss [126]. In a human mesencephalic cell-based model of PD, pre-treatment of the cells with human platelet lysates also protects against ferroptotic cell death [126].

Platelets express also glutamate receptors and exhibit glutamate uptake activity and carry considerable amounts of γ-aminobutyric acid (GABA) in dense granules. Glutamate, the most abundant excitatory neurotransmitter in the brain, and GABA, the major human inhibitory neurotransmitter, are crucial for healthy brain function; abnormalities in glutamate and GABA signal transduction pathways are associated with many neurodegenerative conditions such as PD, Alzheimer’s Disease (AD), and ALS (Figure 4).

Another similarity between platelets and neurons is the production and secretion of amyloid-ß. Platelets are silos of the amyloid precursor protein (APP), containing about 90% of the circulating APP in their plasma membrane and *α*-granules. APP is cleaved by ß-secretase into amyloid-ß and secreted in the blood by activated platelet. APP acts as a platelet receptor and it is involved in thrombosis and coagulation, whereas amyloid-ß promotes platelet aggregation. Amyloid-ß induces platelet activation by binding to the scavenger receptor CD36 and GP1bα and activating the p38 MAPK/COX1 pathways. These pathways induce the release of TXA2, triggering platelet activation, adhesion, and aggregation (Figure 4) [127].

In AD, a neurodegenerative disease that progressively leads to the loss of neurons and consequent dementia, deposition of amyloid-ß in the brain tissue and cerebral vessels is one of the most important neuropathological mechanisms. A recent study has shown that platelets are hyperactive in AD patients and a transgenic mice model of AD. Moreover, the early development of platelet inclusions in cerebral blood vessels in AD mice suggests a role of platelets in amyloid-ß plaque formation. Another work showed that platelets promote the formation of amyloid-ß aggregates in the brain vasculature and that amyloid-ß itself is able to activate platelets [128].

Platelet dysfunction is associated with several other neurodegenerative diseases. In Huntington’s disease, a hereditary autosomal dominant neurodegenerative disorder, platelets display many abnormalities, including aberrant amplification of adenosine A receptor signaling, nitric oxide metabolism dysregulation, and elevated monoamine oxidase activity (MAO). MAO is a mitochondrial enzyme that catalyzes the oxidative deamination of dopamine and presents two different isoforms, A and B, and MAO-B is expressed by platelets Increased platelet MAO-B activity has been positively correlated with Huntington’s disease progression (Figure 4).

Several studies suggest that elevated platelet MAO-B activity has been associated with neuronal damage in many other degenerative conditions, such as PD. In PD patients, many other platelet alterations have been observed, including increased mean platelet volume and decreased glutamate uptake [128].

In conclusion, platelets can regulate neural cells, contribute to brain plasticity and carry pro-neurogenic factors, and multiple mechanisms are involved in platelet-neural cell communication.

For their neuroactive effects, platelets could represent a potential target in neurodegenerative diseases. On the one hand, antiplatelet drugs could be introduced in treatment strategies in order to reduce platelet activity and/or inhibit the overproduction, for example, of amyloid-β in AD patients [129].

Thanks to cell-specific interactions mediated by their receptors, platelets may be used also as a drug delivery system to target specific cells that are difficult to access, such as neurons. For example, platelets have been recently suggested as a model system of glutamate and GABA transport in patients suffering from neurodegenerative conditions [128].

## 4. Platelet Extracellular Vesicles

### 4.1. General Overview of Extracellular Vesicles and Their Clinical Applications

The most recent research in the field of regenerative medicine has brought to light a new interest in the translational and therapeutic potential of extracellular vesicles (EVs). EV is an umbrella term encompassing cell membrane-derived structures differing in their size and including exosomes (30–100 nm), microvesicles (100 nm–1 µm), and apoptotic bodies (>1 µm) [130]. The nomenclature is somewhat confusing and used to be misused [131] until the International Society of Extracellular Vesicles (ISEV) clarified the terminology by reclassifying them into small EVs (sEVs), less than 200 nm in diameter, and medium/large EVs (m/lEVs), more than 200 nm in diameter. Alternatively, the EV density, the biochemical composition, or the isolation technique can be used as additional classification criteria [132]. In general, EVs can be defined as heterogeneous structures delimited by a lipid bilayer and unable to replicate, since they do not contain a functional nucleus [132].

EVs have been implicated in several physiological and pathological functions including immune surveillance, oncogenesis, cardiometabolic, and neurologic disorders and their detailed description is beyond the scope of this review. Here, we provide only a brief overview of three main clinical areas where EVs have a demonstrated involvement. In general, their role can be recapitulated primarily by their ability to mediate intercellular communication [133].

In oncogenesis, EVs have been implied in microRNA transfer between malignant cells to transmit chemotherapeutic resistance [134], or from a malignant to a nonmalignant cell to selectively silence gene expression and induce transformation to cancer cells [135]. Some authors have proposed EV involvement in the metastatic spread, by its harboring molecules necessary for the epithelial-to-mesenchymal transition [136] and/or by acting at a distance and prime a metastatic niche to welcome the migrating metastatic cells appropriately or to locally promote cancer cell growth [137,138]. Regarding their clinical applications, they have been proposed as biomarkers of tumor monitoring [139,140] and as therapeutic vectors for small molecule delivery [141,142].

In cardiology, Emanueli et al. have shown that the concentration of circulating exosomes in plasma correlates with the levels of cardiac troponin, increasing 24 to 48 h after coronary artery bypass surgery [143]. Furthermore, EVs have been implicated in cardiac hypertrophy and remodeling in general, as well as in cardiometabolic disorders where they could mediate vascular damage in metabolic diseases such as diabetes and obesity [144,145].

In neurology, recent studies suggest a role of the EV content in neurodegenerative disorders such as AD or PD, where a correlation between the clinical manifestations and altered expression of miRNA or synaptic proteins has been described in blood and cerebrospinal fluid (CSF) EVs [146,147]. Similarly, a study showed that the circulating levels of EVs containing tau protein, a general marker of neuronal damage, were more elevated in American football players compared with healthy controls, and were probably related to a chronic subclinical traumatic encephalopathy already described in these subjects [148]. Finally, the therapeutic use of EVs has been explored in AD through the use of engineered EVs containing small interfering RNA (siRNA) to alter the expression of beta-amyloid [149] or EVs derived from stem cell-derived EVs to mitigate the clinical manifestations [150,151].

Despite the promises of the different diagnostic and therapeutic applications of EVs, most studies are still preclinical, and clinical use is hindered by the lack of clarity regarding EVs biogenesis, a necessary prerequisite to fully understanding their potential, in the absence of standardization in the isolation and characterization techniques, and limited information about the influence of age, gender, and ethnicity in humans [133].

### 4.2. From “Platelet Dust” to pEV Full Recognition

Platelets are one of the main bodily sources of EVs [152,153]. Their ability to release EVs was already reported in 1967 by Peter Wolf, who coined the expression “platelet dust” to describe the microscopic lipid-rich particulate he obtained after ultracentrifugation of plasma and serum samples [154]. Later on, further studies prompted the evolution of this term to the more accurate one of “platelet microparticles”, which eventually entered the umbrella definition of pEVs according to the ISEV’s recommendations [132]. Surprisingly, the attention of most studies on the therapeutic role of EVs has been focused on ex vivo sources, such as mesenchymal stromal cells (MSCs), with many protocols being already in place in the field of regenerative medicine [155,156]. Nevertheless, using blood cells as a source of EVs would represent a great advantage in terms of cost, manufacturing techniques, and safety. Indeed, ex vivo sources of EVs require at least two steps for their production, i.e., isolation followed by expansion and differentiation in a growth medium, whereas body-cell-derived EVs are already available after collection from autologous or allogeneic donations. Despite the dependence on blood donors, in vivo sources of EVs require little manipulation and eliminate the concerns regarding potential contamination by the growth medium, thus circumventing regulatory issues related to the manufacturing process [157]. Their inherent biocompatibility, compared to MSC-derived EVs or synthetic nanoparticles, also makes them “immunologically transparent” and better tolerated by the recipients [158]. Even more, unlike platelets, pEVs can cross tissue barriers, including the blood-brain barrier (BBB), especially when it is inflamed [159], also thanks to their expression of adhesive receptors such as integrins or Siglec molecules [119,160], thus greatly expanding their use beyond the blood compartment [153].

pEVs bear a phenotype that reflects the typical characteristics of the original platelet, including expression of phosphatidylserine (PS), CD31, CD41, CD42, CD61, CD62, and CD63. Their cargo, too, reflects their parental origin since p-EVs contain growth factors, cytokines, chemokines, lipids, neurotransmitters such as serotonin, and nucleic acids (messenger RNA and microRNA) that mediate a delicate balance between proinflammatory and anti-inflammatory, pro-coagulant and anti-coagulant, and pro-angiogenic and anti-angiogenic events. Furthermore, at least some p-EVs have been found to include mitochondria [161,162].

### 4.3. pEV Isolation, Characterization, and Production Methods

Since pEVs are true biological products, reporting their isolation and characterization techniques used is paramount in getting formal approval as therapeutic agents. Unfortunately, as in the case of PCs, different preparation methodologies have been used without reaching a unanimously shared protocol. Moreover, the isolation technique utilized greatly affects the nature of pEVs [87].

The most common isolation methods are based on ultracentrifugation and density gradient [163] even though size exclusion chromatography, the use of antibody affinity columns, or filtration techniques should be considered as well. A possible hurdle in pEV isolation is the overlap in their size and/or density with other particles such as lipoproteins. Beyond the impact of their purity on the effective number of pEVs obtained, a great number of lipoproteins might also have undesirable biological impacts, since they may behave as proinflammatory effectors [87]. A comparison of different isolation techniques has shown that the best approach is a combination of methods, which decreases the risk of co-isolating lipoproteins but also decreases the pEVs’ yield. In particular, size exclusion chromatography is worth being included in the process, since it is highly efficient in removing free and potentially interfering blood biomolecules. Nevertheless, so far, there has been no proof of therapeutic superiority of either isolation technique [164,165]. An important point is that all these methods require several pre-analytical manipulations representing potentially stressful conditions that might promote platelet damage or alter the pEV’s morphology and functional characteristics [108].

The next step in EV manipulation is their characterization in order to correctly identify their source and to determine their specific phenotype. The most-used techniques are electron microscopy and nanoparticle tracking analysis (NTA) to analyze size and morphology, while Western blot, flow cytometry, and mass spectroscopy are used to identify the expression of surface proteins and the cytosolic cargo [163].

Regarding the source, a recent review [87] identified PRP as the most frequent source used to prepare pEVs. Even though pEVs are already generated under physiological conditions, their use for therapeutic purposes might prompt the need to further enhance their production beyond their constitutive release, and therefore a still-controversial issue is whether platelets should be triggered by an agonist before pEV isolation. Active PCs may perform better not only in quantitative terms (absolute number of pEVs obtained) but also concerning the qualitative properties of pEVs, such as PS exposure on the outer membrane layer or release of growth factors with therapeutic potential [87]. Induction of pEV release can be achieved using several agonists such as collagen, thrombin, ADP, and arachidonic acid as well as calcium ionophores and LPS [157]. Interestingly, the agonist used does not seem to have a significant effect on the pEV’s size. However, in quantitative terms, calcium ionophores seem to ensure the greatest yield, even though also thrombin is a potent inducer, especially if compared with the weaker activity of LPS. Conversely, calcium ionophore-induced pEVs bear a smaller protein cargo, reflecting a poor packaging capacity [166]. On the whole, the choice of the best inducing technique reflects a delicate balance between the absolute yield of pEVs from a single platelet and the amount of cargo and physical properties of the pEVs obtained. In other words, manufacturing techniques should take into account both quantitative and qualitative issues, both of which are greatly influenced by the chosen activation pathway [165,167,168,169] and the storage time before induction of pEV release. Indeed, old stored platelets seem to give a greater yield of pEVs compared to fresh platelets [87].

Storage is another important caveat that must be solved to harmonize pEV processing and preservation techniques. It is noteworthy that, differently from PCs, pEVs can tolerate freezing [170,171,172], thus eliminating the concerns regarding special storage and transport requirements or the need to be used within a few days. Additionally, in this case, the storage temperature may affect both the morphology and function of pEVs [173], but it is not clear whether the best condition is −80 °C or −20 °C [87]. It would be worth examining and comparing different storage conditions, the use of cryopreservatives, and their impact on the viability and therapeutic potential of pEVs.

### 4.4. Therapeutic Applications of pEVs

So far, many therapeutic applications of pEVs have been investigated, even though most studies are still limited to in vitro or animal experiments. The main rationale behind their therapeutic potential is the vast array of molecules expressed on their membrane and their rich content in signaling molecules and growth factors, which allow them to be perfect candidates for intercellular interaction and for acting at a distance on different targets (Table 1 and Figure 2).

Dealing with hemostasis, pEVs have a proven effect on vascular permeability [174] and seem to be even more pro-coagulant than activated platelets [172,175,176]. In the field of regenerative medicine, pEVs might represent a promising approach to wound repair, since the growth factors in their cargo promote fibroblast and keratinocyte migration and proliferation [177,178]. In muscle [179] and bone regeneration, they can enhance stem cell intra-articular engraftment and differentiation [180], as well as promote expansion and prevent apoptosis of local cells, also through the activation of intracellular signaling cascades including the Wnt/β-cateninin [181] or the Akt/Bad/Bcl-2 pathway [182]. In addition, their mitogenic effect could be attributed also to their genetic content (miRNA) [183]. In neurodegenerative disorders, pEVs can promote the proliferation of neural stem cells thanks to their growth factors, more than the use of the same growth factors alone [116,125,128,184]. Their success is also supported by their proven angiogenic potential, primarily derived from their content rich in PDGF, VEGF, and bFGF [184,185,186].

The few studies on human subjects are mainly focused on pEV as an outcome measure, for instance as an index of therapeutic efficacy of antiplatelet drugs (study identification numbers NCT02931045; NCT04578223), or dietary supplements (NCT03203512), or as diagnostic biomarkers (NCT05530330). However, at the time of this review (January 2023), there are a few ongoing or recently completed interventional studies on human subjects using pEVs as a therapeutic approach, in particular for the surgical treatment of chronic tympanic membrane perforations (NCT04761562), the nonsurgical treatment of chronically inflamed post-surgical temporal bone cavities (NCT04281901), the treatment of patients with acute myocardial infarction undergoing percutaneous coronary intervention (NCT04327635), and to promote healing of skin grafts (NCT04664738).

Platelet EVs can be exploited both directly due to their intrinsic properties, and indirectly as delivery vehicles whose cargo can be adjusted according to different therapeutic needs. Their peculiar structure, i.e., an outer lipidic bilayer and an inner aqueous environment, makes them capable of housing both hydrophobic and hydrophilic drugs [157,187]. Platelets may be preloaded with the drugs and then induced to release pEVs using different agonists. Alternatively, the post-loading of already formed pEVs can be chosen. In both cases, loading can be achieved passively through incubation in a drug-containing medium or actively through sonication, electroporation, uptake after saponin treatment or freeze-thaw cycles, and transfection [157].

The most straightforward clinical application of drug-loaded pEVs is probably in the field of oncology, since the interaction between platelets and cancer cells is a consolidated phenomenon, and cancer cells are capable of internalizing pEVs [188,189]. Therefore, pEVs might be used as Trojan horses for anticancer drug delivery. Intriguingly, Michael et al. demonstrated an anticancer effect already for native pEVs by showing the inhibition of lung and colon carcinoma growth in mice transfused with platelet-derived microparticles (PMPs) and identified miR-24 as the main factor responsible for the induction of tumor cell apoptosis in vivo [190]. Furthermore, when synthetic nanoparticles (NPs) are coated with platelet membranes, targeting of tumor tissues is improved, which proves that the molecules displayed on the NP surface are the major determinants of the NP fate [191,192].

A pioneering approach has been tried to facilitate the delivery of antiviral therapies. In an in vitro study, pEVs entrapping the anti-HIV drugs lamivudine and tenofovir comparatively increased the inhibitory effects on HIV-1 replication, while decreasing cytotoxicity, likely due to the slower release by pEVs [193]. Evidence of pEV implication in viral disease emerged during the COVID-19 pandemic with findings of higher levels of pEVs in hospitalized SARS-CoV-2-positive patients compared with uninfected hospitalized controls [108].

Finally, the pEV role in cardiovascular disease is also well established [187]. pEVs have been shown to correlate with the size of the myocardium at risk after an acute coronary syndrome (ACS) [194] and to act as biomarkers of vascular inflammation, perhaps because they contain some proinflammatory isoforms of C-reactive protein (CRP) [187]. This also explains their putative involvement in atherogenesis, where pEVs may interact with endothelial cells and leukocytes, acting as functional bridges to mediate monocyte recruitment to the vascular walls [195], and induce macrophage apoptosis [196], which contributes to the formation of foam cells. Surprisingly, pEVs have not been found within the atherosclerotic plaque itself but only in the blood of atherosclerotic patients. Atherosclerotic plaques contain microvesicles, but not of platelet origin [197], which instills the doubt that they might be simple bystanders of the atherosclerotic process instead of active players in its pathogenesis [187]. Nevertheless, the fact that they can be engineered to modify their cargo and used as drug-delivery systems has inspired new therapeutic platforms, such as the one developed by Pawlowski and colleagues, where pEV-like NPs loaded with a thrombolytic drug were successful in obtaining targeted fibrinolysis in preclinical models [198]. In addition, not all pEVs are associated with cardiovascular risk. Indeed, also natural pEVs have shown therapeutic potential. For instance, transfusion of pEVs from rats after hind limb ischemia-reperfusion conditioning into rats with middle cerebral artery occlusion was able to reduce the infarct area [199], and a similar result was obtained in rat models of limb ischemia [200], suggesting that the transfer of conditioned or “educated” pEVs might be protective and partially reverse the injury already occurred. The precise mechanism underlying this cardiovascular protective role is still poorly understood but likely derives from the rich biological cargo of pEVs as well as from their capability to deliver it to several cell types in a finely targeted manner [187].

In summary, from their humble origin as “platelet dust”, pEVs have certainly made huge steps forward, and they have been established not only as biomarkers with diagnostic, prognostic, and predictive significance but also as promising therapeutic strategies.

## 5. Therapeutic Hitchhiking: Platelets as Drug Delivery Vehicles

If until now the discussion has been focused on platelet-derived biomaterials and fragments such as extracellular vesicles as a therapeutic opportunity, it would also be worth considering the potential ability of platelets themselves to become drug delivery vehicles. Indeed, their multifaceted characteristics make them good candidates for therapeutic drug delivery. First, they are the second-most abundant blood cell types after erythrocytes, and they are easier and quicker to purify [201]. Second, their biological properties allow them to encapsulate both hydrophobic and hydrophilic drugs [157] and to store them mainly in their open canalicular system [159]. In this way, they can cloak the drug and hide it from the body, decreasing its clearance rate and meanwhile allowing a more targeted delivery of the drug with fewer systemic effects [159]. Third, their biocompatibility allows them to travel through biological structures almost undisturbed and with little or no immunogenicity [201]. The best example is probably their ability to cross the BBB [159].

The fact that platelets release the encapsulated drug only when they are activated and undergo degranulation [159] allows us to define their mechanism of targeted drug delivery as a true deceit for the body. For instance, in the field of oncology, tumor cell-induced platelet aggregation can be exploited to let platelets release the chemotherapeutic drug directly into the tumor mass. In other words, a process that normally would contribute to tumorigenesis can be converted into a therapeutic attempt at the expense of the tumor itself, as has already been demonstrated for doxorubicin-loaded platelets in adenocarcinoma [202] or lymphoma cell lines [203]. Similarly, in the case of occluding strokes, where platelet aggregation plays a pivotal pathological role at the site of an unstable atherosclerotic plaque, they have been hypothesized as effective carriers of an anticoagulant drug. The rationale behind this promising hypothesis is that, since platelets will likely be activated at the ischemic site, this will be also the site where the drug will be released, ensuring a targeted therapy [159].

Several steps forward have been made in the field of platelet engineering, i.e., a method of modifying platelets through chemical and/or physical mechanisms favoring their engulfing of the desired drug, coating them with biomolecules, or changing their properties to improve their drug-delivery ability. One of them consists in exploiting the abundant thiol and amine residues on their membrane to chemically link them to biomolecules [201]. Another technique is electroporation, which allows molecules to penetrate the cell through pores created in its membrane [204]. For example, Rao et al. have used this method to load platelets with nanorods to facilitate phototherapy in the treatment of head and neck squamous cell carcinoma in mice. Conveying light energy through an external irradiating source, the nanorods converted it into heat energy, injuring the tumor cells [205].

The limited lifespan of platelets and their relatively low stability in response to external stimuli have prompted the need to think about valid alternatives. The advent of biomimetic platelets, i.e., artificial platelets that mimic the biological properties of their natural counterparts, represents a recent achievement in biomedical engineering [206]. In practice, biomimetic platelets must bear at least three key platelet properties: flexibility, discoid morphology, and ability to aggregate upon activation. This can be achieved by using bi-layered nanoparticles coated with platelet-mimicking peptides [207]. Nevertheless, platelet-mimicking biomaterials eliminate the greatest advantage of their natural counterparts, i.e., the evasion of the immune system. Being synthetic, they are characterized by high immunogenicity, and they often have lower biological efficacy. For these reasons, they are still not used in clinical practice [201].

There are still many unanswered questions that must be investigated, not only in terms of posology issues (e.g., the amount of drug encapsulated by the platelet, the platelet concentration to be administered, the best timing and way of administration) but also referring to the possible, and probably unavoidable, alteration of platelet characteristics induced by the drug itself. To cite just one of these concerns, it cannot be excluded that an anticoagulant drug taken up by a platelet to treat stroke will alter platelet features, for instance by changing its morphology, reducing its capability as a drug delivery vehicle, or increasing the immunogenicity. Furthermore, the release of the drug precisely at the desired site might still seem utopian [159], and further in vivo studies are deemed necessary to better disentangle the several doubts that remain in this fascinating but still pioneering area of research.

## 6. Conclusions

Platelets are multifunctional blood components, capable of changing shape and function, responding rapidly to environmental changes, and fulfilling distinct context-dependent functions throughout life, just as did Proteus.Activated platelets release numerous bioactive molecules from α-granules, δ-granules, and lysosomes. The platelet secretome influences many physiological and pathophysiological processes beyond hemostasis, such as inflammation, immunity, neurogenesis, and oncogenesis. Platelets also exhibit complex interactions with many different cells, beyond endothelial immune cells in the circulation [1].

Based on their multifaceted functions and multiple cell interactions, we assume that the platelet population is probably composed of subgroups with different functions: hemostatic platelets, immune platelets, sentinel platelets, helper platelets, and scavenger platelets. We intend to test this hypothesis in the future. Certainly, the in-depth knowledge of all their functions will help to understand the pathogenesis of numerous pathologies and to reveal new therapeutic properties with pleiotropic effects.

Nowadays, antiplatelet drugs are the first-choice therapy for the treatment of cardiovascular disease and the prevention of atherothrombosis. However, growing evidence indicates that antiplatelet drugs are also effective in cancer treatment strategies and neurodegenerative disease therapy. Interestingly, not only drugs targeting platelet function but also platelets themselves show potential as therapeutic strategies. Attention has also been focused on the use of platelet concentrates in regenerative medicine and pEVs as a new approach to develop functional nanoparticles for disease-targeted delivery. Many clinical trials have been carried out in the last two decades, mainly in the fields of orthopedics, maxillofacial surgery and dentistry, ophthalmology, and cardiac surgery. Furthermore, despite being at its onset, the study of platelets as drug delivery vehicles is showing promising results, which are predominantly derived from high biocompatibility, versatility and minimal immunogenicity.

Platelets, apparently simple cellular fragments, hide a world, mostly still unknown, with enormous diagnostic and therapeutic potential, now summarized in this review.

## Figures and Tables

**Figure 1 ijms-24-04565-f001:**
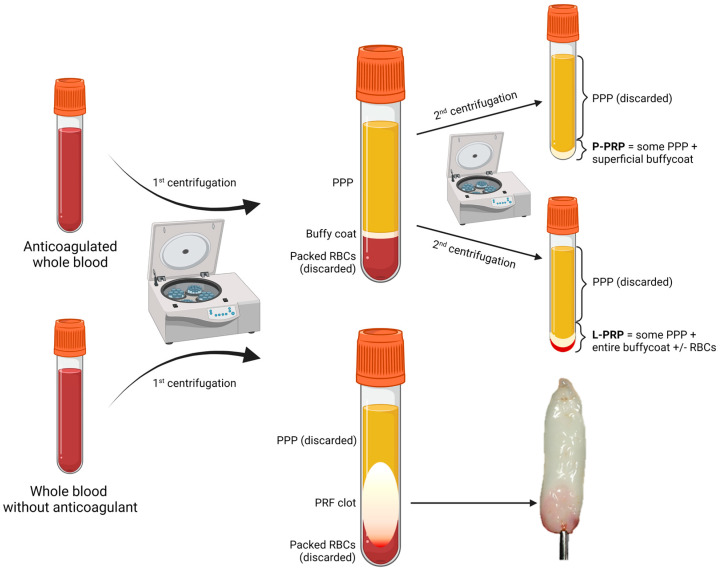
Platelet concentrate processing techniques. The upper part of the figure shows the preparation of P-PRP and L-PRP. The centrifugation steps allow the separation into PPP, buffy coat, and packed RBCs. The main difference between P-PRP and L-PRP derives from the amount of buffy coat collected between the first and the second centrifugation steps. In the lower part of the figure, PRF preparation is displayed: after blood collection in tubes without anticoagulant, a flexible and malleable fibrin clot is obtained through centrifugation. Created in BioRender.com (accessed on 17 Fabruary 2023). Abbreviations: L-PRP, leukocyte and platelet-rich plasma; PPP, platelet-poor plasma; P-PRP, pure platelet-rich plasma; RBC, red blood cell.

**Figure 2 ijms-24-04565-f002:**
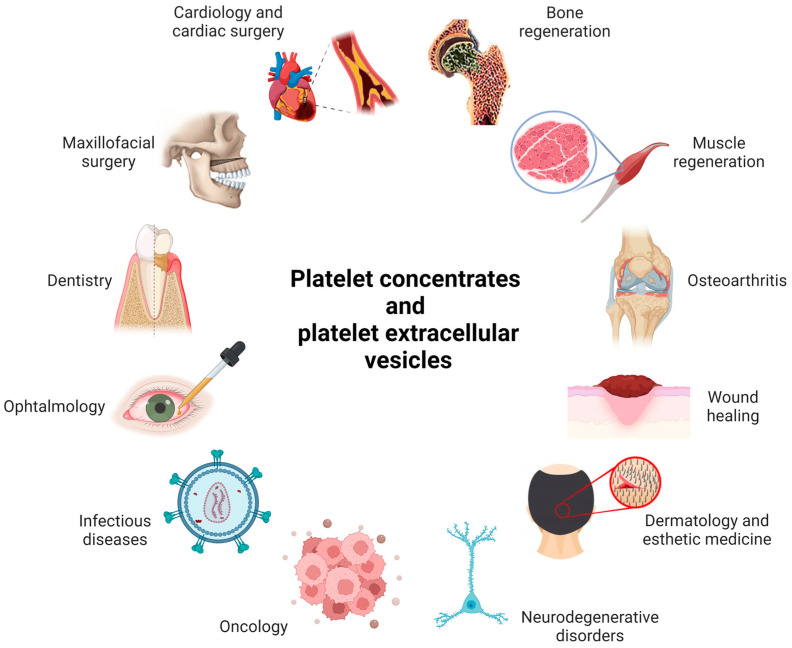
Main applications of platelet derivatives. Created in BioRender.com (accessed on 16 January 2023).

**Figure 3 ijms-24-04565-f003:**
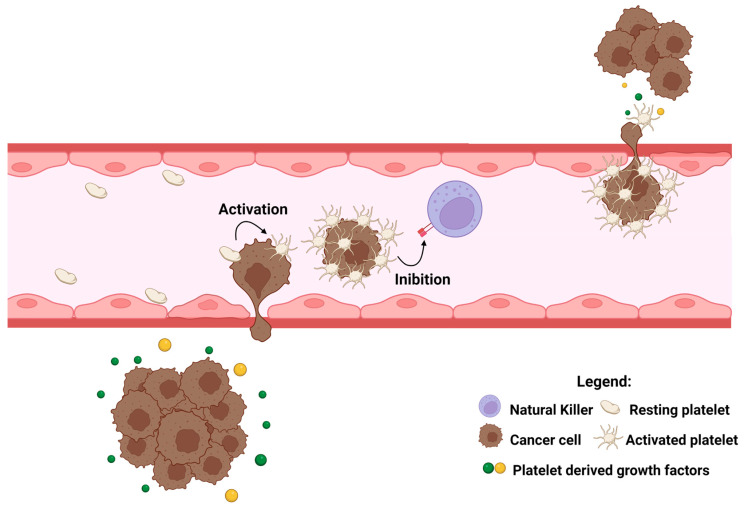
Cancer-platelet interplay. The cancer-platelet interaction is bidirectional. Cancer cells express several molecules able to interact with platelet receptors capable of forming aggregates of platelets and cancer cells, which leads to platelet activation and creation of a pro-thrombotic environment on one side, and to tumor cell extravasation on the other. In turn, platelet activation by cancer cells promotes the release of the platelet secretome, including proinflammatory cytokines and growth factors that promote tumor growth, modulate the tumor microenvironment by recruiting and activating leukocytes, and favor the formation of metastases. Furthermore, platelet-derived TGF-β decreases the NK cell patrolling activity, thus increasing tumor cell survival. Created in BioRender.com (accessed on 29 December 2022).

**Figure 4 ijms-24-04565-f004:**
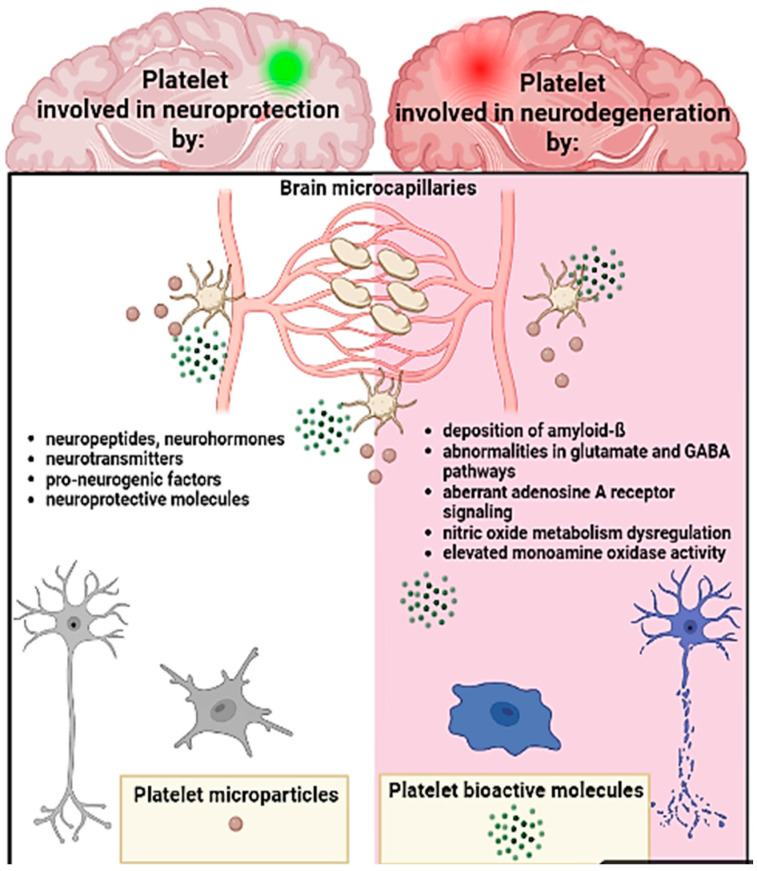
Platelet role in healthy brain function and association with neurodegenerative disorders. The left side of the figure shows, the role of platelets in brain physiological processes, including their modulation of neurogenesis and synaptogenesis through the release of bioactive molecules contained in their granules or platelet-derived exosomes and microparticles, their bridging function between the blood and the brain, and their contribution to neuronal communication through the release of neurotransmitters. In contrast, the right side displays their involvement in neurodegenerative processes such as their putative role in β-amyloid plaque formation in AD or metabolic dysregulation associated with Huntington’s disease. Created in BioRender.com (accessed on 12 January 2023).

**Table 1 ijms-24-04565-t001:** Growth factors released by platelets or identified in platelet derivatives.

Growth Factor	Source	Function	Refs.
**BDNF**	Platelets, neurons, heart, lung, liver, skeletal muscle	MSC survival and proliferation, angiogenesis, neuron activity, survival, and function	[40,41,42]
**b-FGF**	Platelets, macrophages, T lymphocytes, mast cells, endothelial cells, fibroblasts, chondrocytes, osteoblasts, mesenchymal cells	Mitogenic effect on fibroblasts, endothelial cells, MSCs, chondroblasts, and osteoblasts, angiogenesis, bone regeneration, corneal tissue repair, hair growth, wound healing, soft dental tissue regeneration, embryogenesis	[34,35,37,39,40,43]
**BMP**	Platelets, bone, CNS and many other tissues	Immune system regulation, cell maturation and differentiation, angiogenesis, cartilage and bone formation, fracture repair, tooth development, tissue fibrosis, cancer cell inhibition, CNS function	[37,40]
**CTGF**	Platelets after endocytosis from bone marrow extracellular environment	Found in platelets at a concentration that is 20-fold higher than any other PGF, involved in platelet adhesion, angiogenesis, cartilage regeneration, fibrosis, platelet adhesion, white blood cell migration, angiogenesis, regulation of collagen synthesis	[34,35,37,38,39]
**EGF**	Platelets, macrophages, monocytes	Endothelial chemotaxis, epithelial and mesenchymal mitogenesis, regulation of collagenase secretion, keratinocyte and fibroblast migration and proliferation, angiogenesis, re-epithelialization, corneal tissue repair	[34,35,37,39,40,43]
**IGF-1**	Platelets, epithelial cells, endothelial cells, fibroblasts, smooth muscle cells, osteoblasts	Differentiation of osteoblasts in bone and of myeloblastic tissue in muscle, proliferation, migration, angiogenesis, neuroprotection and re-myelination, bone regeneration, regulation of collagen production, synergistic effect with PDGF	[39,40,43]
**PDGF**	Platelets, endothelial cells, macrophages, monocytes, smooth muscle cells, osteoblasts, keratinocytes	First growth factor to be released in a wound, mitogenesis, chemotaxis of macrophages and neutrophils, regulation of collagen synthesis and collagenase activity, matrix formation and remodeling, angiogenesis, neurogenesis, bone regeneration, re-epithelialization, wound healing, corneal tissue repair, synergistic effect with TGF-β	[34,35,37,39,40,43]
**TGF (α-β)**	Platelets, activated Th1 cells, NK cells, monocytes, macrophages, endothelial cells, neutrophils, keratinocytes, fibroblasts, muscle cells, bone and cartilage ECM	MSC proliferation, mitogenesis of fibroblasts, osteoblasts, and endothelial cells, chemotaxis of endothelial cells, angiogenesis, regulation of collagen synthesis and collagenase secretion, extracellular matrix formation and connective tissue regeneration, regulation of mitogenesis mediated by other growth factors, bone formation and regeneration, re-epithelialization, wound healing, cancer metastasis, inhibition of macrophage and lymphocyte proliferation, synergistic effect with PDGF, possible anti-proliferative effect at high concentrations	[34,35,37,39,40,43]
**VEGF**	Platelets, endothelial cells, fibroblasts, keratinocytes	Differentiation and mitogenesis of endothelial cells, chemotaxis, angiogenesis, increased vascular permeability, vascularization, neuroprotection, bone regeneration	[34,35,37,39,40,43]

**Abbreviations**: BDNF, brain-derived neutrophic factor; BMP, bone morphogenic protein; CTGF, connective tissue growth factor; ECM, extracellular matrix; EGF, epidermal growth factor; FGF, fibroblast growth factor; IGF-1, insulin-like growth factor-1; MSC, mesenchymal stem cell; NK, natural killer; PDGF, platelet-derived growth factor; TGF, transforming growth factor, Th1, T-helper 1; VEGF, vascular endothelial growth factor.

**Table 2 ijms-24-04565-t002:** Sources, technical preparation, and characteristics of platelet concentrates.

Product	Source	Preparation Steps	Characteristics	Refs.
**P-PRP** **L-PRP**	ACD-anticoagulated whole blood or apheresis	1. Short soft-spin centrifugation to separate PPP, BC, and packed RBCs; 2. Collection of PPP and BC; 3. Long hard-spin centrifugation and discarding of the supernatant (PPP) to obtain P-PRP or L-PRP.	The main difference between P-PRP and L-PRP consists in the amount of BC collected between the first and the second centrifugation steps, i.e., only the superficial layer for P-PRP and the entire buffy coat (possibly with some residual RBC) for the L-PRP. Temperature-sensitive (PRP cannot be frozen).	[35,48,49]
**PRF**	ACD-anticoagulated whole blood or apheresis	Immediate centrifugation to obtain a flexible fibrin clot.	Slow and dynamic polymerization allowing a more flexible, high-density, and stable fibrin scaffold. Harder but malleable biomaterial, easily adaptable to anatomical surfaces with valuable applications in ENT (ear, nose, and throat), maxillofacial, and plastic surgery. Enrichment with VEGF and TGF-β compared with PRP.	[37,48,50,51,52,53,54,55]
**PG**	ACD-anticoagulated whole blood or apheresis	Same preparation as P-PRP or L-PRP but additional activation step to form a semi-solid (gelated) product by the action of calcium chloride or calcium gluconate with or without the addition of human thrombin, batroxobin, or synthetic TRAP (a thrombin receptor agonist), often added directly at the application site.	Low-density, almost liquid biomaterial, lacking a true support matrix but easier to apply locally. Use of bovine thrombin associated with the development of antibodies against coagulation factors II, V, and XI leading to life-threatening coagulopathies. Versatile kinetics of growth factor release depending on the activator used (faster when using thrombin).	[35,37,56,57,58,59]
**PL**	Whole blood or apheresis platelets	Platelet degranulation after freeze-thaw cycles, sonication, treatment with solvents and detergents, possibly followed by activation by thrombin or batroxobin just before application.	Cost-effective. Easier preservation (compared with PRP, it can be kept for 14 days at 4–10 °C or longer if frozen).	[40]
**E-S**	Whole blood	Clotting at room temperature for 2–72 h, centrifugation to obtain serum and dilution with saline solution.	Nonallergic. Biomechanical and biochemical properties similar to normal tears but with a greater content of TGF-β, vitamin A, lysozyme, and fibronectin.	[37,60,61]
**E-PRP**	Whole blood or apheresis platelets	Platelet degranulation after freeze-thaw cycles, dilution of the supernatant with saline solution.	Higher content of PGFs in a smaller volume, more suitable for intraocular instillation.	[37,56,62]

**Abbreviations**: ACD, acid/citrate/dextrose; BC, buffy coat; E-PRP, platelet-rich plasma eye drops; E-S, serum eye drops; PG, platelet gel; PGF, platelet growth factor; PL, platelet lysate; P-PRF, pure platelet-rich fibrin; L-PRF, leukocyte and platelet-rich fibrin; L-PRP, leukocyte and platelet-rich plasma; P-PRP, pure platelet-rich plasma; RBC, red blood cell; TGF-β, transforming growth factor-β.

## Data Availability

No new data were created or analyzed in this study. Data sharing is not applicable to this article.

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
