# Peer review of "Platelets, Protean Cells with All-Around Functions and Multifaceted Pharmacological Applications"

_ijms, 2023, doi:10.3390/ijms24054565_

Round 1
Reviewer 1 Report
I would like to recommend this manuscript for publication after minor revision:
1. Generally speaking, a manuscript has 3-5 key words, but if the 7 key words in this article are appropriate, please simplify.
2. Interventional therapy may lead to platelet aggregation and activation, which may lead to thrombosis <(1)Zhijin Han, Haojie Guo, Yifan Zhou, Liguo Wang, Kun Zhang, Jing-an Li, Composite coating prepared with ferulic acid to improve the corrosion resistance and blood compatibility of magnesium alloy, Metals 12 (2022) 545. (2)Jingan Li, Wei Li, Dan Zou, Fang Kou, Yachen Hou, Aqeela Yasin, Kun Zhang, Comparison of conjugating chondroitin sulfate A and B on amine-rich surface: for deeper understanding on directing cardiovascular cells fate, Composites Part B: Engineering 228 (2022) 109430.>. The authors are invited to make a brief discussion in combination with the literatures.
3. The reference format of the article does not conform to the MDPI standard, please modify it carefully according to the template.
4. What is the repetition rate of this review article?
Author Response
REPLY TO REVIEWER 1
- Generally speaking, a manuscript has 3-5 key words, but if the 7 key words in this article are appropriate, please simplify.
We reduced the number of key words from 7 to 4, eliminating “atherosclerosis”, “neurodegenerative disease”, and “cancer” since they refer to more specific paragraphs of the review. We kept “platelets”, “inflammation”, “platelet derivatives”, and “extracellular vesicles” to refer to the more general topics considered in this work.
- Interventional therapy may lead to platelet aggregation and activation, which may lead to thrombosis. The authors are invited to make a brief discussion in combination with the literatures.
A new paragraph has been added to the section “Platelets and atherosclerosis”, referring to platelet involvement in thrombosis after intravascular stent placement and to the strategies used in the field of biomedical engineering to overcome this risk using biocompatible devices that prevent platelet aggregation, favor re-epithelialization, and reduce the risk of restenosis. The authors are grateful to the Reviewer for the recommendation of the articles by Han et al. and Li et al., which have been cited in the added paragraph.
- The reference format of the article does not conform to the MDPI standard, please modify it carefully according to the template.
The reference format has been modified using square brackets to cite in the text and the format “Author, Title, abbreviated Journal name, Year, pages” at the end of the manuscript. Using the Mendeley software, a compromise in the style choice was necessary. The Authors hope that this new format will better conform to the MDPI standard.
- What is the repetition rate of this review article?
We calculated the repetition rate using this formula [(1- number of distinct words / number of words) x 100] and this online tool (https://www.repetition-detector.com/?p=online) to find the repeated words. Considering the whole text without the abstract and the Table and Figure captions, we counted 8800 distinct words (total words – repeated words = 10566-1766). The repetition rate is thus (1 - 8800 / 10566) * 100 = 16,71%.

Reviewer 2 Report
This review focuses on the plentiful role of platelets in different diseases and the pharmacological applications of platelets and derivatives. I found the review to be an overall well-written review with a lot of information, however, the review needs rephrasing for better interpretation and the authors should emphasize more on the uniqueness and novelty of the review either in the introduction or conclusion section.
Comments:
1) In the Introduction section, the authors should write more about the structure of platelets, receptors, and types of granules.
2) All the figure legends need to be more explanatory and similar information can be omitted from the text.
3) It would be better if the Platelet derivatives section go up after the introduction followed by the role of platelets in diseases. A table summarizing the mechanisms by which platelets contribute to different diseases' manifestations would be beneficial for readers.
4) Section 3, Platelets, and cancer is confusing with many small paragraphs and needs to be rewritten.
5) The authors should also mention the role of platelets as delivery vehicles.
6) References missing from many sentences all throughout. Quote, wherever necessary.
7) The platelet derivatives section can be improved by adding a table conveying all relevant and qualitative information rather than text.
8) Add references in all the tables.
Author Response
REPLY TO REVIEWER 2
1) In the Introduction section, the authors should write more about the structure of platelets, receptors, and types of granules.
Following the Reviewer’s advice, the Introduction section has been explanded with new sub-sections specifically dedicated to platelet granules and receptors.
2) All the figure legends need to be more explanatory and similar information can be omitted from the text.
The figure legends have been expanded with a more detailed explanation of what the image displays in addition to the figure title, so that each element of the picture is commented in the caption for an easier interpretation by the reader.
3) It would be better if the Platelet derivatives section go up after the introduction followed by the role of platelets in diseases. A table summarizing the mechanisms by which platelets contribute to different diseases' manifestations would be beneficial for readers.
Following the Reviewer’s advice, the “Platelet derivatives” section has been moved up immediateley after the introduction, and a new section titled “Platelet role in disease” has been added thereafter including “Platelets and atherosclerosis”, “Platelets and cancer”, and “Platelets, the bain, and neurodegenerative conditions”.
A table summarizing the mechanisms of platelet contribution to pathological processes and the mediators involved has been added after the paragraph dedicated to neurodegenerative disorders. The Authors thank the Reviewer for the suggestion.
4) Section 3, Platelets, and cancer is confusing with many small paragraphs and needs to be rewritten.
Section 3, “Platelets and cancer”, has been entirely rewritten. Paragraphs have been enriched and new citations from the literature have been added.
5) The authors should also mention the role of platelets as delivery vehicles.
Following the Reviewer’s request, an entire paragraph (“5. Therapeutic hitchhiking: platelets as drug delivery vehicles”) has been added after the section dedicate to platelet extracellular vesicles and just before the conclusion.
6) References missing from many sentences all throughout. Quote, wherever necessary.
New references have been added throughout the text and in the Tables.
7) The platelet derivatives section can be improved by adding a table conveying all relevant and qualitative information rather than text.
According to the Reviewer’s advice, the “Platelet derivatives” section has been shortened by substituting the rather long explanation of the different platelet concentrates with an easier-to-understand table summarizing the preparation steps and the peculiar characteristics of each. The deleted text has been substituted by the sentence “The processing steps and the peculiar characteristics of each PC are presented in detail in Table 2, while Figure 3 shows a cartoon exemplifying the technical preparation of P-PRP, L-PRP, and PRF underlying the differences between the three.”
However, if the Reviewer thinks the deleted part is worth keeping in addition to the new Table, the Authors will restore it as it was in the first manuscript version.
Figure 1 (previously Figure 3) of this section has been slightly modified to improve its comprehension by the reader.
8) Add references in all the tables.
A new “Ref.” Column has been added to both Table 1 and Table 2 to introduce the references. In the Supplementary Table, no “Ref.” column has been added since the studies mentioned in each row can be easily identified through the NCT number and the link to the website of ClinicalTrials.gov.

Round 2
Reviewer 2 Report
The revised review looks good to me. The authors answered all the comments and the article can be accepted.